# Automation of System Security Vulnerabilities Detection Using Open-Source Software

João Pedro Seara *  and Carlos Serrão

Information Sciences, Technologies and Architecture Research Center (ISTAR), Lisbon University Institute (ISCTE-IUL), 1600-189 Lisbon, Portugal; carlos.serrao@iscte-iul.pt
* Correspondence: joao_pedro_seara@iscte-iul.pt

**Abstract:** Cybersecurity failures have become increasingly detrimental to organizations worldwide, impacting their finances, operations, and reputation. This issue is worsened by the scarcity of cybersecurity professionals. Moreover, the specialization required for cybersecurity expertise is both costly and time-consuming. In light of these challenges, this study has concentrated on automating cybersecurity processes, particularly those pertaining to continuous vulnerability detection. A cybersecurity vulnerability scanner was developed, which is freely available to the community and does not necessitate any prior expertise from the operator. The effectiveness of this tool was evaluated by IT companies and systems engineers, some of whom had no background in cybersecurity. The findings indicate that the scanner proved to be efficient, precise, and easy to use. It assisted the operators in safeguarding their systems in an automated fashion, as part of their security audit strategy.

**Keywords:** systems security; cybersecurity; vulnerabilities; scanner; open-source software; automation

## 1. Introduction

Occurrences related to cybersecurity are frequently highlighted in the media. Over the years, there has been a noticeable increase in global cyberattacks [1]. A study conducted by IBM had some interesting key findings [2]. The average cost of a single breach amounted to USD 4.45 million to organizations in 2023. This represented a 2.3% increase from the previous year's 4.35 million. Going further back, the total cost in 2020 was 3.86 million, so a consistent increase has been observed. Also, this study concluded that cost saving from high levels of DevSecOps adoption amounted to USD 1.68 million per organization. Future projections, from a different study, indicate a sharp increase in the overall global expenditures related to cybersecurity incidents, estimating the global cost to reach USD 10.5 trillion annually by 2025 [3]. This works out to a roughly USD 28.8 billion daily cost, or USD 333 thousand each second.

The challenge of providing organizations with cybersecurity expertise extends beyond simply finding security professionals; it also involves finding professionals who possess the necessary experience. As a result, hiring becomes a complex task [4,5].

Underprivileged nations are particularly susceptible to the challenges described above, due to their insufficient cybersecurity infrastructure, lack of inter-agency coordination and emergency responses, limited Information and Communication Technology (ICT) skills, and inadequate protection of critical national infrastructure [6].

Combating the aforementioned problems can be aided by automating security audits. Automated technologies offer a methodical approach to these audits and eliminate the need for a knowledge ramp-up. The author of [7] supports this statement, by pointing out that these automated systems will serve as the "cornerstones of cyber defense strategies". Some other authors [8] go further and forecast that automation tools are merely a first step

towards what they refer to as "cyber autonomy", a state wherein defense systems will be completely autonomous.

In relation to Artificial Intelligence (AI), it should be noted that Large Language Models (LLM) and other AI tools are rapidly gaining attention and adoption worldwide. Automated cybersecurity mechanisms and AI are already interacting, as automation outputs can be fed into AI algorithms that use data sets to cross-check them and then determine the best course of remediation action [9].

Automating security auditing involves some important considerations. It is crucial that the automation process assigns different severity levels to different security vulnerabilities. This aspect holds significant importance in ensuring the proper prioritization of the overall security measures [10]. Also, it is essential to highlight that adhering to current standards, policies, and guidelines, necessitates a systematic approach to cybersecurity auditing. An exemplary framework in this regard is ISO 27001/27002 [11].

From all the previous paragraphs, one can conclude that security incidents have a negative impact on organizations in various ways. The negative impacts of security incidents on organizations go beyond just the loss of productivity, media attention, and damage to reputation, as these incidents also result in significant financial costs for organizations. Getting the right professionals takes time and is expensive. All these problems are exacerbated in organizations from developing countries. In order to mitigate these risks and continue their operations, organizations are often required to adhere to cybersecurity regulations, and employ a systematic approach to cybersecurity auditing (namely, proper prioritization of vulnerability fixes).

It was also asserted above that, to address these challenges, automation plays a significant role in facilitating security audits. Automated tools and technologies can streamline the auditing process, making it more efficient and effective. These tools can scan networks, systems, and applications, for vulnerabilities. They can identify potential threats, prioritize different vulnerabilities, and provide recommendations for remediation. By analyzing data and identifying critical vulnerabilities or high-risk areas, organizations can allocate their resources and efforts more efficiently. This ensures that the most pressing security issues are addressed promptly, reducing the risk of security incidents and their associated impacts.

The objective of this research is to address the aforementioned problems and requirements by creating an artifact that performs systematic security auditing and many of the auxiliary tasks that revolve around it (which usually require a specialized engineer). It aims to enhance the ongoing efforts of academic institutions and businesses in automating security audits, while also making the findings accessible to the wider community.

The produced artifact is a cost-effective, comprehensive, and user-friendly vulnerability scanning solution, comprising a different set of modules. The first module of this scanner is the *Network Discovery* module. This module is responsible for scanning and identifying all devices connected to a network, including computers, servers, routers, and IoT devices. It utilizes advanced scanning techniques to detect open ports, services, and vulnerabilities present on each device. The second module of the scanner is the *Vulnerability Assessment* module. This module performs in-depth scans of each device identified in the network discovery phase. It utilizes a comprehensive vulnerability database to identify known vulnerabilities and misconfigurations on each device. This module also provides recommendations and remediation steps to address the identified vulnerabilities, helping users improve the security posture of their network. The third module of the scanner is the *Reporting* module. This module generates detailed reports summarizing the findings of the Network Discovery and Vulnerability Assessment modules. The reports include information such as the devices scanned, vulnerabilities found, severity levels, and links to recommended actions. Figure 1 shows a graphical representation of the relationship between these modules (this image will be further explained and detailed in Section 3).

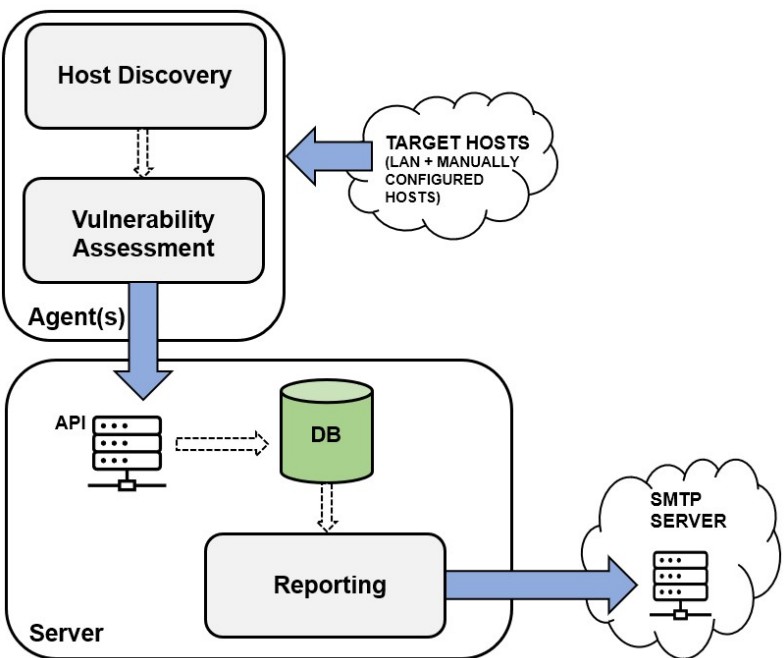

**Figure 1.** Graphical representation of the relationship between this artifact's modules (arrows depict data flow).

It also important to notice that this work incorporates emerging DevSecOps principles, such as the amalgamation of security-focused technologies known as Security Orchestration, Automation, and Response (SOAR) [12].

This document begins with a brief introduction to the topic at hand. Then follows a description of the methods and conclusions of other relevant solutions already attempted before, to prove the relevance of the work being presented. The next part discusses the design considerations and the implementation of the resulting artifact. The following section will present the results of the validation tests, which were carried out both internally and with the assistance of external testers. The final portion will elaborate on the results accomplished in this work, and suggestions for future developments.

## 2. Related Work

The first stage of the research was searching for previously attempted solutions to the problems defined above.

In their scholarly work, the authors of [13] have meticulously engineered an advanced security auditing framework employing the capabilities of both the *libnet* and *libpcap* libraries. Libnet, the first of the two libraries, furnishes developers with a powerful API that facilitates the crafting and dispatching of network packets across a variety of protocols. This capability is crucial for simulating network traffic that mimics various interactions and behaviors within a network infrastructure. Complementarily, libpcap stands as the second pivotal library. It is tasked with the interception and meticulous examination of network packets in real time. The aforementioned solution excels in its capacity to rigorously identify and document active network ports, which are potential entry points for unauthorized access if left unsecured. Moreover, by scrutinizing network interactions, the framework can infer the operating system details of the host machines. Additionally, the solution integrates an assessment of system vulnerabilities by cross-referencing findings with the Common Vulnerabilities and Exposures Database (CVE DB). This integration enables the artifact to pinpoint known security flaws that could be exploited by malicious entities. This advanced level of analysis, coupled with the ability to simulate networking events and monitor traffic in such a granular manner, makes the security auditing tool developed by the authors an asset for organizations seeking to bolster their cyber defense mechanisms.

In [14], the authors have developed a vulnerability scanner in *Python* called "Net-Nirikshak". It performs the enumeration of targets (finding open ports and running services) and interfaces with NVD to detect known vulnerabilities associated with them. In addition to this, it also identifies and reports SQL injection vulnerabilities, highlighting all the susceptible links on the target. Moreover, the tool possesses the ability to exploit these SQLI-vulnerable links and retrieve confidential information from the target. The tool automatically generates a report, which is then sent to a designated list of email addresses.

In [15], the implementation of a vulnerability scanner based on NVTs is described. NVTs are network vulnerability test plug-ins provided by *OpenVAS/Nessus*, and they are known for their vast quantity and daily update service. By combining multiple NVTs, vulnerabilities can be detected effectively. The authors conducted a thorough analysis of network-based vulnerability scanning and the usage of NVTs. Based on this analysis, they designed a network vulnerability scanning system that relies on them. They also performed a test of this system and presented its potential applications.

The work in [16] describes a vulnerability scanner that focuses on web servers, especially SQL injection and XSS. It performs vulnerability assessment by using *pocsuite3*, an open-source vulnerability scanning framework for web services. It has a web interface for the end user. During the research project, four primary examinations were conducted: information collection testing, port scanning testing, SQL injection detection testing, and XSS detection testing.

The authors of [17] have developed a vulnerability scanner that also focuses on web servers. This tool detects a comprehensive range of vulnerabilities, including cross-site scripting attacks, SQL injection, and directory traversal. No specific information is available regarding the user interface. Although it supports the generation of reports, continuous auditing is not supported.

The authors of [18] have developed a vulnerability scanner called "FalconEye", which also focuses on web servers. This artifact has an interesting design aspect: the scanning process is distributed across servers that act as "workers". It uses common messaging protocols such as AMQP to handle the communication between the components. The primary objective of this system is to identify vulnerabilities (CVEs) specifically associated with web applications, such as XSS and XXE injection. There is no information regarding the user interface. The FalconEye system comprises three modules: an input source module, a scanner module, and a support platform module. The input module enhances the scope of the target server, while the other modules enable the system to conduct comprehensive scanning for generic vulnerabilities. The authors assert that the results substantiate the system's potential to be a valid competitor among the numerous detection systems currently available.

The work in [19] implements a vulnerability scanner based on *Nmap* that supports target enumeration, vulnerability scanning, and remote network mapping, and is aimed at organizations and professionals who have little expertise in cybersecurity (a scope that is similar to this work's). It has a web-based interface. The authors argue that the arduous process of acquiring proficiency in multiple command line tools, along with their intricate functionalities and parameters, can be circumvented by utilizing advancements like this tool. Consequently, this has the potential to broaden the availability of security testing, especially for small and medium-sized enterprises.

The artifact developed in [20] is called "SecuBat". It focuses on web applications, has a website crawler and tests the target for SQL injection and XSS vulnerabilities. It has a graphical user interface and an API that allows the user to launch custom attacks. It implements reporting (but not continuous auditing) and stores historical data. The researchers state that, by utilizing this particular tool, they successfully identified numerous websites that could potentially be vulnerable. In order to validate its precision, they selected one hundred websites from the list of potential victims for in-depth examination. Consequently, they confirmed the existence of exploitable weaknesses in the identified web pages, which included prominent multinational corporations and even a finance

ministry. Over fifty entities responded, either by providing supplementary information, or by notifying that the security vulnerability had been addressed and resolved.

Finally, the work in [21] describes another high-level scanner that focuses on web vulnerabilities. It performs a URL crawl and attacks the resulting URLs to detect XSS, CSRF, SQL injection, and other vulnerabilities. It has a web interface allowing it to start scans, and it generates a report at the end. The authors assert that through the utilization of the tool on various websites, they have identified a total of 45 vulnerabilities that can be exploited, including XSS, SQL injection, directory disclosure, as well as local and remote file inclusion.

None of the solutions above are prepared for automatic network detection. Memory exhaustion issues have been reported with some of them.

From the analysis of all the tools described above, the following can be concluded:

- Some of the solutions are too narrow-scoped. In other words, they are not of a general nature. They focus either exclusively on specific operating systems or on specific services;
- Most of the solutions require network access rights to access the target hosts when running from outside their network, with the exception of the solutions that allow the installation of local agents. Some of them also require the credentials of the target host to be known;
- Most solutions do not automatically determine the infrastructure information. This means that the information about the target hosts must be obtained from the system administrators and configured manually in the tool before the scans are executed;
- Some of these scanners require a daemon to be running. This can lead to problems if a fault occurs, e.g., if the hard disk is full or the memory is exhausted, and therefore requires the implementation of a watchdog that restarts the service if necessary. The PnP philosophy is not adopted, as no tool or framework has been found that is immediately ready to use. Quite the contrary—all require prior configuration before execution.

The fact that all solutions examined have at least one of these shortcomings means that no examined artifact solves the problem that this work attempts to address. The work presented in this paper differs from these projects, in that it synthesizes a number of features that address the current shortcomings into a single open-source artifact that requires little to no skilled personnel to install and operate. Further details are given in the next section regarding architecture and implementation.

## 3. System Design and Implementation

The initial part of this section presents a comprehensive outline of the initial design decisions that were undertaken to ensure the achievement of the intended objectives. Subsequently, it delves into the specifics of the artifact's architecture, the selection of appropriate tools, and the step-by-step process of its implementation.

### 3.1. Design Choices

The primary objective of this project was to create a vulnerability scanning and reporting system that is accessible to all users without requiring any expertise in cybersecurity. In order to achieve this goal, a set of specific characteristics, or sub-goals, were established for the solution:

- *Agent-server architecture*—This choice enables effortless scalability (through the addition or removal of agents) and the adaptability of deploying agents directly within LANs. This placement behind firewalls and proxies allows for the direct targeting of host systems. Meanwhile, the operator can conveniently access all metrics from all agents through a single server. Additionally, it is feasible to access metrics from agents without relying on a server, granting the operator the ability to possess a portable vulnerability scanner. Further elaboration on these aspects will be provided subsequently;

- *Low-cost hardware*—The primary emphasis of the agent hardware is on the *Raspberry Pi* board, which uses the ARM architecture. However, the solution should possess the adaptability to function on a conventional computer/server with an x86 architecture;
- *Free software*—The implementation must solely rely on open-source software (FOSS) and tools, with a preference for those that are portable and have minimal resource requirements to accommodate the hardware's limitations. The selected options for this purpose include *Linux* as the operating system, Python as the programming language, Nmap as the port scanning tool, and *MongoDB* as the database;
- *Scalability*—Adding and removing processing power to/from the solution should be a straightforward task. The proposed method involves employing a multi-agent solution, which allows for the seamless addition or removal of agents within the environment.
- *Modularity*—The flexibility of implementation enables users to enhance and personalize their experience as much as possible. By adopting an API-centric approach, users have the freedom to directly engage with the system through the standard CLI or create their own clients, web frontends, mobile applications, or AI/ML systems for data manipulation and the generation of remediation proposals. Moreover, the agents can function independently from the backend server, allowing for standalone usage, if necessary;
- *Plug and play (PnP)*—After installation, the system should be operational and ready to use. However, it should also offer a high level of customization, allowing the operator to configure it according to specific requirements. The operator has the option to define which scans to execute and which hosts to target. In the absence of operator intervention, the application should automatically identify and scan hosts within its vicinity. Being "plug and play" implies that the agent–server communication should be established using a minimal number of ports; more specifically, only HTTP/HTTPS. Moreover, this communication should only occur in an outbound manner, with the agent initiating contact with the server. This design choice facilitates the functioning of agents in environments with firewalls and restricted network access, which usually allow for outbound connections by default for well-known ports. Furthermore, the process of discovering vulnerabilities and conducting scans should be continuous and ongoing. It should operate continuously and iteratively in the background without requiring any human intervention. This ensures that the system remains vigilant and up-to-date in identifying potential security risks;
- *E-mail reporting*—Configured recipients should receive automatic e-mails containing security vulnerability reports. Additionally, the level of detail in these reports should be adjustable, allowing for the filtering of vulnerabilities that are exploitable [22]. This feature is crucial as it enables the identification and prioritization of vulnerabilities that require immediate attention and resolution;
- *Security*—Communication between the various components of the solution and the end users must undergo authentication and be conducted securely over the HTTPS protocol, which uses SSL/TLS encryption;
- *Future-proofing*—The progression of the technological landscape, specifically in terms of IPv6 capability, should be taken into account.

In conclusion, the created artifact not only exploits the existing cutting-edge technology, but also addresses certain deficiencies that were previously highlighted. Its implementation aspects will now be detailed.

### 3.2. Implementation

The developed work comprises two fundamental software components, namely, the *Agent* and the *Server*. A high-level overview of the architecture is illustrated in Figure 2, showcasing a diagram of the system.

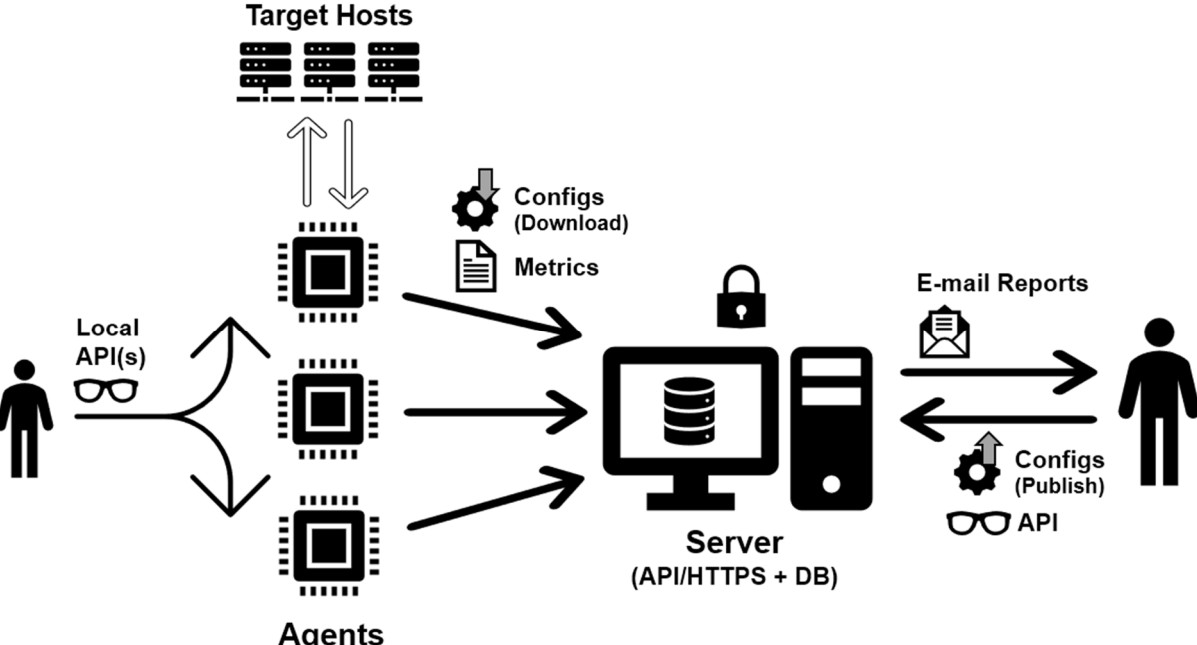

**Figure 2.** High-level architecture.

The diagram's central portion depicts a multitude of agents, which can be any Linux machine or VM. However, in accordance with the proposed paradigm in this study, it is assumed that these agents are Raspberry Pi boards. These agents establish contact with the server, with the arrows indicating that the connections are initiated from the agents towards the server. All communications between the agents and the server employ HTTPS and HTTP password authentication, ensuring security.

Previously, it was mentioned that this work is divided into three main modules. We will now dive down into each one of them.

3.2.1. Network Discovery Module

This module runs in the agents. Before performing the actual vulnerability assessment, the agents need to gather a list of targets to scan, from both the neighborhood and a manually defined list given by the operator. This module's operation can be divided into three main functions, which are now described below.

The first function looks at the operating system's tables to locate ARP (IPv4) and NDP (IPv6) entries. These tables store details about Layer 2 hosts in the neighborhood, specifically the IP/MAC address pairs that are already known to the agent's underlying OS. To access this information, the agent utilizes the *ip neigh* command at the OS level. The function disregards any local link addresses or reserved ranges. Additionally, in the case of IPv6, it excludes all addresses except for Global Unicast addresses, so as to prevent the interference of transient addresses present in the network.

The second function performs automatic discovery. It involves the utilization of the *Scapy* module for Python to send ARP requests to all the IP addresses within the local IPv4 subnets that each of the network interfaces belongs to. This process aims to identify available hosts in the vicinity by recording positive ARP replies. However, only IPv4 address spaces with a subnet mask larger than/16 are scanned, to prevent an overwhelming number of addresses that would require an impractical amount of time to scan (exceeding 50,000 addresses). Similarly, the probing of IPv6 subnet spaces is omitted due to their vast size. For example, the standard/64 subnet space, which represents the smallest locally usable IPv6 subnet, encompasses over 18 quintillion addresses.

The third and last function will parse and use the contents of a string of comma-separated IPs (IPv4 or IPv6) and/or domain names. This string is a configuration field

that can be modified by the operator. In cases in which the input is a domain name, the function will try to resolve it into its IP address(es) using the system's configured DNS. If it cannot be resolved, it is ignored. In cases when the target host happens to be in the local network, the MAC address is obtained using ARP and then added to the host's record. Unlike automatic discovery, this function adds the hosts to the list of neighbors, whether they are online or offline.

Figure 3 shows a graphical representation of the process described above.

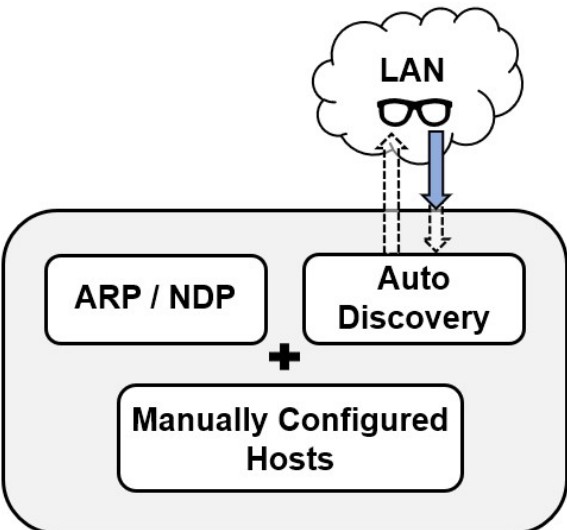

**Figure 3.** Graphical representation of the Network Discovery module (arrows depict data flow).

Finally, the module will try to ping the found hosts, and also try to find their DNS names by performing a reverse lookup. This list will be used by the agents to perform an Nmap vulnerability scan, more specifically by the Vulnerability Assessment module, which is now detailed below.

3.2.2. Vulnerability Assessment Module

As with the previous one, this module runs in the agents. This module performs a scan on each of the members of the list of target hosts obtained before by the Network Discovery module. The scanning process consists of three parts: the first part involves identifying the running operating system on the target host, and the running services on all open UDP and TCP ports; the second part entails running a vulnerability scan using Nmap scripts against those ports; the last part consists in parsing this output into a comprehensible dictionary (Python object) of vulnerabilities found per host.

Going into further detail, the module iteration starts by grabbing the list of the hosts to target from the outputs of the Network Discovery module. Then, this list is used to launch multiple parallel workers—one per target host—and, inside each of these workers, enumeration, scanning, and output parsing are performed.

Enumeration is performed by invoking the OS detection function from the *python3-nmap* module, together with an option to scan open ports (TCP and UDP). In its turn, this function leverages the operating system's Nmap command-line with the flags that enable OS detection and also the detection of open ports and running services. When this function is triggered, the most common 1000 ports are probed in order to check their state. The banners received in reply are used to determine which service is running on each of these ports. When it comes to the OS detection, the agent—again using Nmap—sends a specially constructed packet towards the target hosts and analyzes the reply, and then tries to guess the most probable OSes that the target might be using, based on the specifications of the packet received (for example, different OSes use different TCP headers, making them uniquely identifiable).

At the end of the enumeration function, the module will have collected a list of hosts, their OSes, and open ports/services. The next step is performing a vulnerability scan against each of those ports.

A scan is triggered for all ports detected in the hosts, sequentially iterating all ports. If more than one script is configured in the agent's configuration, then multiple scans (one for each script) are sequentially triggered during each port's iteration. This vulnerability scan is done by leveraging the capability of Nmap to run NSE scripts that automate such tasks (more details about this capability below). The command is called, once again, via the python3-nmap module. The list of scripts to run can be configured by the operator, and these can be a single script name, or a category of scripts. By default, the "vuln" category is enabled.

At the end of the vulnerability scanning function, the module will have collected the information from the enumeration function, plus a list of vulnerabilities found per script, for each of the ports.

The next step is parsing the vulnerability scanning outputs. This needs to be done as python3-module returns the outputs in the XML format, so it must be parsed.

While the output is parsed, a dictionary of vulnerabilities is created for each of the scanned ports, using the CVE ID of the vulnerabilities (or any other vulnerability ID, if the DB is different) as keys. The value of these keys is a list of strings, containing the CVSS (severity score), description, and a related link that has more information about that specific vulnerability (and possibly remediation information as well). Plus, if the vulnerability is exploitable, a tag is also added to this list, to make it easily identifiable (this tag will be used later by the artifact's logic if, for example, the operator configures the sent reports to contain only exploitable vulnerabilities).

The parser detects whether a vulnerability is exploitable, depending on whether at least one of these is positive: if the word "exploit" is part of the DB name (for example, the script "vulscan" organizes the outputs per vulnerability DB, and some of these DBs are exclusive to exploits); if the word "exploit" is part of the vulnerability name or ID; and finally, if this same word exists in the description of the vulnerability.

Finally, statistical information is added to the data being collected: the number of ports scanned, the number of scripts ran, the number of vulnerabilities found, the number of exploits found, and the total duration of the scan.

All the data collected in the paragraphs above are merged into a single dictionary object for this target host, and then this object is returned to the main loop by the forked worker, right before it ends its job.

The last step of the module iteration is waiting until all forked workers end. Once all the forks end, all the outputs are merged into a single dictionary object, which is now ready to be uploaded to the server's API.

Figure 4 shows a graphical representation of this process.

The system administrator has the ability to configure scanning options. Some configuration examples instruct the scanner to only consider manually configured hosts, or to restrict scanning to a list of network ports, or even change the list of scripts that the scanner should run.

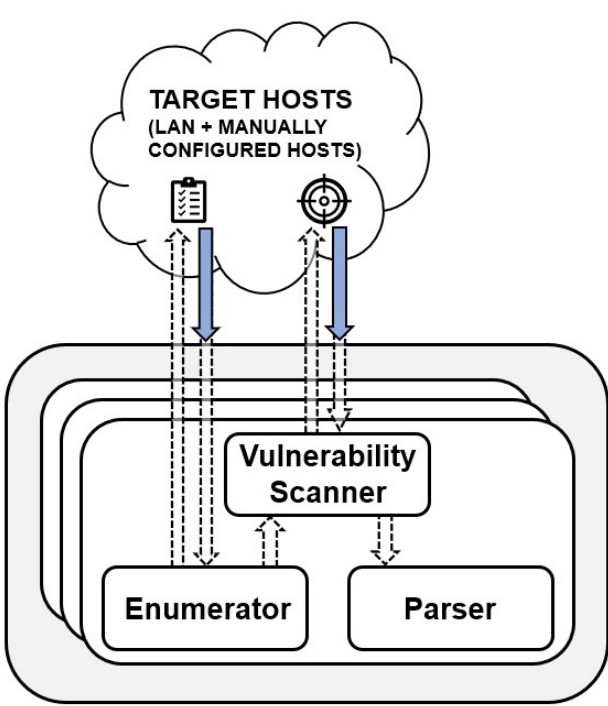

**Figure 4.** Graphical representation of the Vulnerability Assessment module (arrows depict data flow).

3.2.3. Agent–Server (and Client) Communications

All the metrics collected by the agent are consolidated into a single JSON object and periodically transmitted to the server. Figure 5 provides a visual representation of the structure of this JSON object.

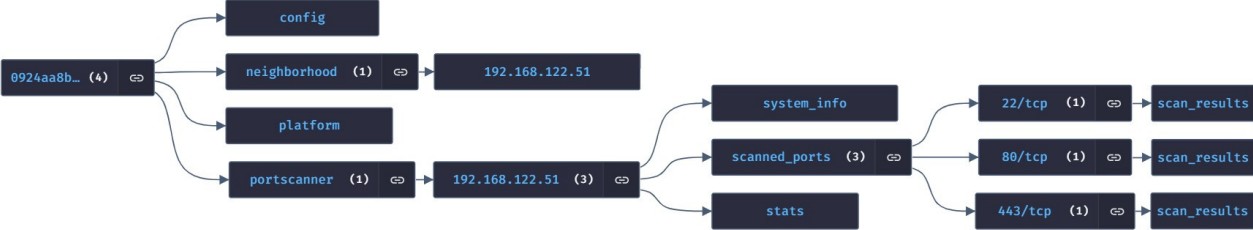

**Figure 5.** Graphical representation of the JSON object's schema.

Looking at the figure above, one can assess that, under the root key (which is the agent's unique ID, allowing the server to distinguish it from other agents), four main keys can be observed. The first key displays the presently active configuration values of the agent (some of these have already been described above). The second key exhibits the IP addresses of the hosts discovered within the agent's vicinity, exemplified by 192.168.122.51 in this instance (These data are collected by the Network Discovery module, already detailed above). The third key encompasses information concerning the agent's own hardware and OS, such as CPU, storage, and memory usage. Lastly, the fourth key encompasses system information, a list of services, results from vulnerability scanning, and statistics of the targeted hosts (data collected by the Vulnerability Assessment module, already detailed above).

Following the transmission of this information to the server during the upload phase, the agents proceed to download the configurations that have been published for them, also from the server. These configurations may be broadcasted to all agents, or specifically tailored for a particular agent UID (this UID is derived from the agent's product or serial number; in cases where this is not feasible, a randomly generated UUID is utilized). These configurations are then implemented locally and become effective immediately. A few

examples of agent configurations could include the scanning frequency, the list of recipients and the granularity of reports to send, or, as already mentioned before, Nmap scripts to be executed, or a list of manually specified target hosts for scanning.

As previously mentioned, the server's API serves as a means for agents to write metrics into, and retrieve their configurations from. This API enables not only the agents but also other clients to access and manipulate data within the server's database (like checking agents' findings or changing agents' configurations). These clients may include the CLI developed in this project, a command-line HTTP client like *cURL*, a web browser such as *Firefox* (which incorporates a JSON parser to present the API data in a human-readable format, as depicted in Section 4.1.2), or any potential frontend that may be developed in the future. Even an AI system can utilize the API's outputs for further analysis.

Furthermore, the API also facilitates the retrieval of historical metric data from the agents. The duration of retention of these historical data is adjustable, and the server employs this value to periodically remove older records from the database.

### 3.2.4. Reporting Module

The server provides support for sending vulnerability scanning reports through e-mail. This module will read, from the server's database, all the metrics collected by all agents. Then, it will parse all of these results into a dictionary, which is then converted into a CSV file, which is finally attached to the e-mail body to be sent.

The operator has to specify a list of recipients and the connection details that enable the server to connect to the SMTP server. Figure 6 shows a graphical representation of this process.

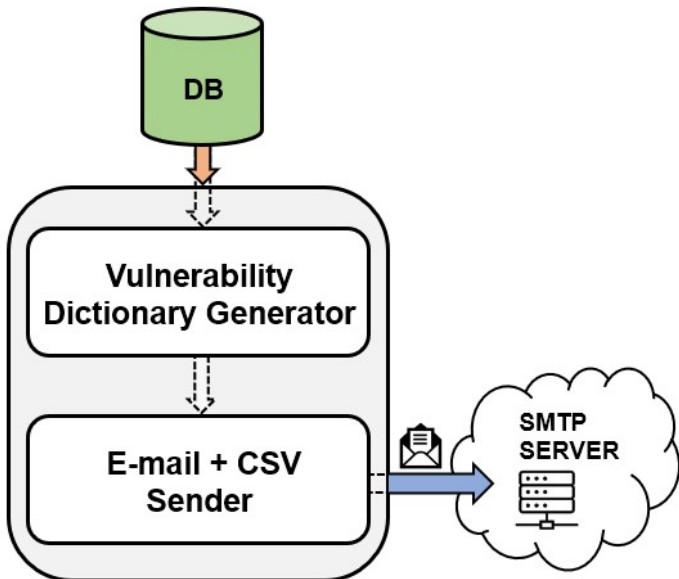

**Figure 6.** Graphical representation of the Reporting module (arrows depict data flow).

The level of detail in these reports can also be adjusted according to the operator's preferences. The reports can include all vulnerabilities, or only focus on the exploits that require immediate attention. This feature proves to be highly beneficial for system administrators, as it enables them to prioritize their actions effectively.

### 3.2.5. Other Considerations

One advantageous aspect of the architecture design is that the data collected by the agents can be accessed directly for read-only purposes, independently from the server. A local API, which is disabled by default, can be activated within the agents. This allows the operator to carry a Raspberry Pi device and conduct vulnerability scans in an isolated

environment, even when the server is not accessible. This can be done by editing the configuration file and then restarting the services, within the agent itself. However, it is important to note that this approach has certain drawbacks. Firstly, the data collected will not be uploaded to the server, resulting in a lack of centralized availability. Additionally, e-mails will not be sent, as this task is performed by the server.

Another possibility is to have an all-in-one (AIO) setup, simply by installing both the agent and the server software packages on a single machine, be it physical or virtual. This configuration provides the operator with a comprehensive vulnerability scanner in a portable VM. However, opting for this setup means sacrificing the flexibility and scalability advantages that come with a distributed architecture using multiple agents.

Moving on to the specific choices made for the software platforms and tools used in the deployment of the artifact, Nmap [23] was selected as the port/vulnerability scanner. Nmap is a straightforward yet powerful tool for vulnerability scanning. One of its advantages is that it does not require a daemon to function. Additionally, it is portable, allowing for easy installation and immediate use through a single installation command. Nmap also supports parallelization, enabling the simultaneous scanning of multiple targets, which proves useful in scanning multiple targets concurrently. Moreover, Nmap supports NSE scripts, which are written in the Lua language. This feature empowers Nmap to perform tasks such as enumeration, vulnerability scanning, and penetration testing (commonly known as *pentesting*).

In addition to the previously mentioned OpenVAS and Nessus, the systematic research conducted also identified other commercial alternatives to Nmap, namely, *Nexpose*, *Scanner-VS*, *Cybot*, *Xspider*, and *Qualys*, as referenced in [24–26]. Commercial tools were also found, such as *Faraday*. However, these alternatives were not considered for use in the presented work. This decision was based on the fact that they either exhibited one of the issues outlined in the concluding part of Section 2, or they required payment, which did not align with the paradigm of this study.

While vulnerability scanning formed the central focus of this work, it was necessary to incorporate several other features. The selection of tools to implement these features did not follow a systematic research approach. Instead, an online search was conducted to identify the most renowned tools that fulfilled each requirement. These tools were then briefly studied, before a decision was made regarding their suitability. The decision was made to utilize Python 3 as the programming language for this project. Python is a widely used high-level programming language that is freely available. It boasts a large and active community of developers and maintainers worldwide. One of the key advantages of Python is its seamless integration with the underlying operating system, allowing for easy file manipulation, retrieval of platform information [27], and execution of commands in the OS shell. Additionally, Python offers an extensive collection of libraries and modules that are essential for implementing the required functionalities. Among these libraries and modules, Scapy stands out as a packet manipulation library. It proves to be particularly valuable for performing operations related to ARP, such as scanning the network adapter's neighborhood. This functionality is especially useful for automating the scanning process of neighboring hosts. A project referenced in [28] demonstrates the implementation of a network scanner that utilizes Scapy to identify hosts within the same subnet.

In relation to the server-side implementation of the REST API, Python offers various solutions. Among these solutions, *Django*, *Flask*, and *FastAPI* were considered. Django is known for its versatility and complexity, which can make it challenging to learn. On the other hand, Flask and FastAPI both possess the necessary functionalities to implement the API effectively. Despite FastAPI having a smaller community and consequently less support, Flask was ultimately chosen as the preferred option. The server API operates behind a reverse proxy Apache web server, which is responsible for authenticating and encrypting HTTP connections between the agents/clients and the API. This is achieved through the utilization of SSL and HTTP authentication modules. Alternatively, Nginx could also be considered as a suitable option. Both of these servers are readily available in

the repositories of most Linux distributions. In this particular case, Apache was selected due to the authors' familiarity with it, as the specific advantages and disadvantages of each web server do not significantly impact the current work.

Regarding the choice of database technology on the server side, the main contenders were *MySQL* and MongoDB. These solutions differ fundamentally, with MySQL being a structured database, while MongoDB is a document-oriented database, commonly referred to as a *NoSQL* database. MySQL ensures greater data integrity as it adheres to the fixed structure of SQL tables. On the other hand, MongoDB is better suited for real-time analytics and offers seamless integration with Python. It readily recognizes Python objects, such as dictionaries, and allows them to be directly uploaded to MongoDB as database documents. Both databases have robust support for Python clients. Ultimately, the flexibility and ease of integration with Python data structures led to the selection of MongoDB.

As part of this project, a Command Line Interface, or just the *CLI*, was created to simplify various tasks such as data consultation or agent configuration management. The implementation of the CLI was facilitated by utilizing *Click*, a specialized Python module for CLI development. In Figure 7, two instances of CLI outputs are depicted, showcasing the help menu output describing the complete range of available commands and a vulnerability report specifically highlighting exploits.

**Figure 7.** CLI showing help and vulnerability report outputs.

As observed previously, the CLI commands produce output in a JSON format that is color-coded. This feature enables operators to utilize a JSON parser such as *jq* to manipulate and filter the output. If the output is redirected, the CLI automatically disables the colorization. This is because colorization introduces special characters to the output, which may not be recognized by JSON parsers.

The development process primarily took place on *Ubuntu* 20.04 "Focal". However, it was also installed and tested on other operating systems, such as *Ubuntu* 22.04 "Jammy" (server edition, or ARM edition in the agents), *Debian* 11 "Bullseye" (main edition, or ARM edition in the agents), and *Raspberry Pi OS* (previously known as "Raspbian") 11 "Bullseye" (exclusively in the agents). These operating systems were chosen as they are the most mature releases available at the current date. Additionally, all the necessary software packages were either present in their native repositories or made available by the respective product owners. It is worth noting that all the aforementioned operating systems are Debian-based releases. Therefore, the installation scripts and source code remain consistent across all of them. Furthermore, Python 3 is readily accessible on all these operating systems.

## 4. Validation

The artifact underwent validation and testing in both a local laboratory and by external testers. The external testers evaluated the artifact in a real-world setting, and subsequently completed a survey. Local tests were conducted to assess technical metrics such as the reliability, accuracy, and security of the agent. On the other hand, the surveys aimed to evaluate usability and quantify the value the artifact adds to an organization.

In terms of local tests, the evaluation of "reliability" focused on two main parameters: service stability over an extended period of uptime, typically spanning several days, and resource usage, specifically CPU and memory, when subjected to stress. The assessment of "accuracy" involved testing the scanner's capability to precisely identify the operating system and service information of the target hosts, as well as detecting any existing vulnerabilities. This was accomplished by creating a controlled test environment with known vulnerabilities and verifying that the scanner produced accurate results. Lastly, the "security" tests aimed to validate that the API does not permit unauthenticated or insecure connections at the protocol or service level.

In an effort to ensure a diverse range of perspectives, the process of selecting external testers aimed to encompass individuals and organizations with varying levels of knowledge and organizational complexity. Consequently, the chosen testers consisted of an IT and cloud business organization, a telecommunications business organization specializing in cybersecurity, an organization operating at the intersection of IT and financial sectors, and an independent IT freelancer with expertise in the open-source field. Further information about these testers can be found in the acknowledgments section at the conclusion of this paper.

An individual tester from each organization was provided with the artifact, along with instructions on how to set it up. These instructions can be found within the project deliverables, the links of which are available at the end of this document. Subsequently, a survey was created, comprising 10 statements pertaining to user experience, 5 statements concerning organizational impact, 2 statements regarding overall experience, and three open-ended questions. The initial 17 statements required the tester to utilize the Likert scale, selecting a value ranging from 1 to 5, in order to express their level of agreement or disagreement with each statement. The surveys were conducted via video call, allowing the testers the opportunity to elaborate on each point, if necessary, and enabling the conversation to encompass different perspectives and opinions on current deficiencies and potential enhancements. Although testers were also given the option to respond in written form, none of them chose this method.

### 4.1. Local Tests

For the local tests, the hardware employed comprised a Sony *Vaio* E11 laptop (2013) functioning as the server/agent with the hostname "JP-OLD". Additionally, two Raspberry Pi devices were utilized as agents, with "RPI4" being a fourth generation Model B (2019) and "RPI1" being a first generation Model B (2012). Furthermore, a virtual machine

named "AIO" was employed for development and testing activities, running on an external hypervisor.

Figure 8 shows a photo of the setup of the local laboratory.

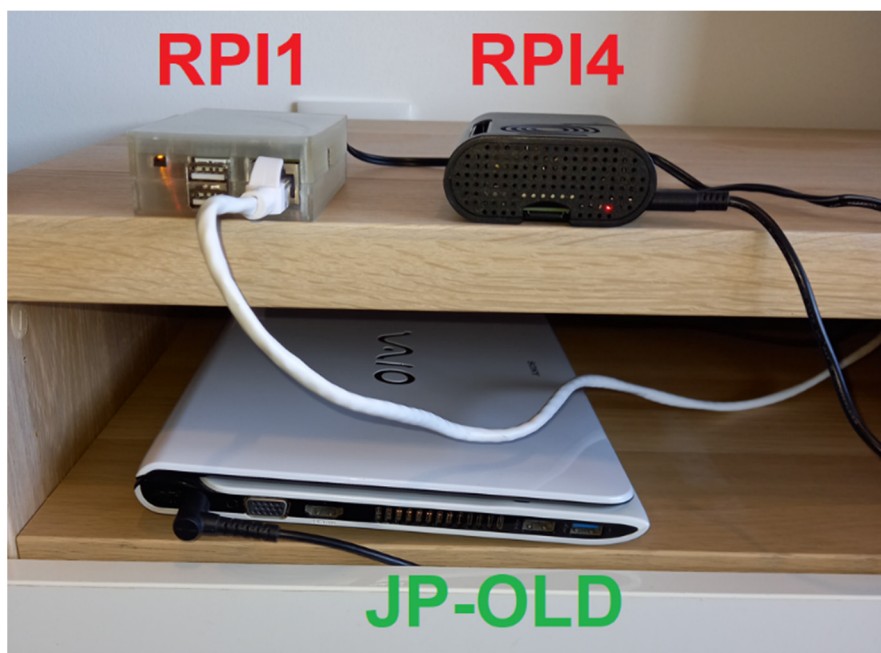

**Figure 8.** Local laboratory.

Specifications of the environment:

- JP-OLD (server and agent, for testing/staging)—1.75 GHz dual-core processor, 8 GB RAM, connected via Wi-Fi;
- RPI4 (agent for testing/staging)—1.5 GHz quad-core processor, 2 GB RAM, connected via Wi-Fi;
- RPI1 (agent for testing/staging)—700 MHz single-core processor, 256 MB RAM, connected via Ethernet;
- AIO (VM running in an external hypervisor; mostly for deployment)—1.8 GHz quad-core processor, 8 GB RAM, connected via Ethernet and Wi-Fi.

4.1.1. Reliability Tests

When it comes to the uptime, no special tests were performed, as the staging environment was always on during the entire deployment period, since the first working builds. This means at least 5 months of mostly continuous hardware uptime with no issues (from November 2022 until March 2023). Service restarts were performed from time to time to test new builds. System logs (link available at the end of this document) show at least 14 days of continuous uninterrupted service uptime in both the server (JP-OLD) and one of the agents (RPI4) between service restarts.

When it comes to testing resource usage, all the staging agents (RPI1, RPI4, and JP-OLD) were left running on default configurations for 72 h, and then data were grabbed for at least the last 24. The same was done for the server running in JP-OLD. This means they were finding and scanning all hosts in the neighborhood (the other agents, the server, an Internet gateway, and any other devices eventually connected to the Wi-Fi network), plus two manual hosts that were configured ("sapo.pt,google.pt"). Three scripts were activated in the configuration ("vuln,vulscan,discovery") (two of these were script categories, so the number of individual scripts was higher than three). To further stress the agents, the local APIs and debug logs were activated, and the time interval between modules' iterations was set to only 5 s.

Figure 9 shows the CPU usage observed in the agent boards, during an entire day of testing, using the tool *Munin*.

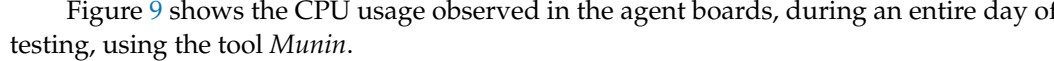

**Figure 9.** CPU usage in the agent boards during tests.

Figure 10 shows the memory usage observed in the agent boards, during an entire day of testing, using the same tool as above.

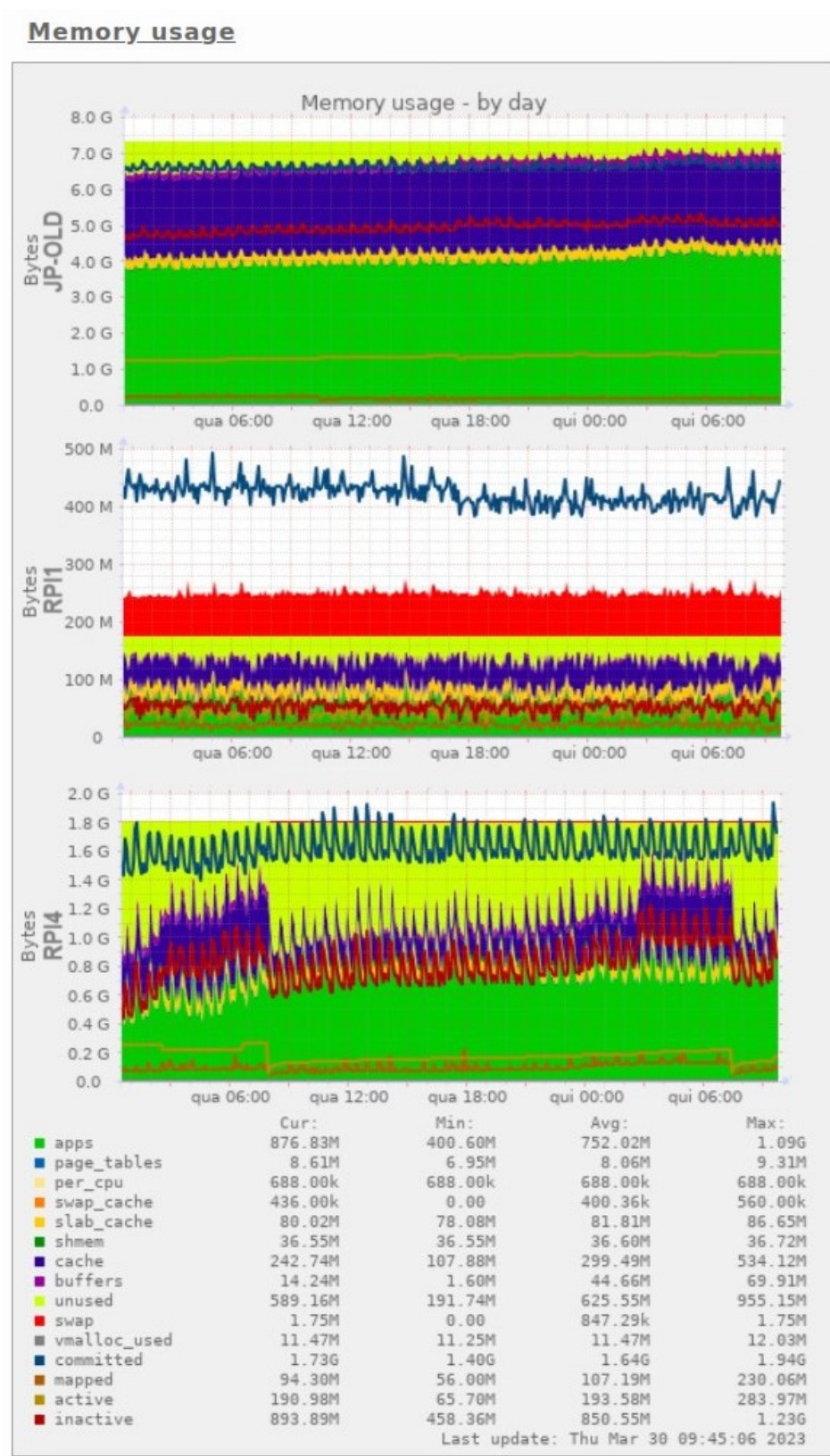

**Figure 10.** Memory usage in the agent boards during tests.

It should be pointed out that, in the case of the agent operating on older hardware (RPI1), certain adjustments had to be made regarding the selection of scripts to execute and the number of workers to initiate. This is understandable considering that the initial versions of Raspberry Pi possess limited resources in comparison to more recent hardware models. Nevertheless, once the appropriate configuration was established, this agent also operated flawlessly for consecutive hours until the conclusion of the tests. In conclusion, the agents consistently and effectively transmitted data to the server throughout the entire duration of the testing period.

4.1.2. Accuracy Tests

The developed artifact demonstrated commendable accuracy in vulnerability scanning by successfully identifying the operating system, services, versions, and vulnerabilities. Although there were occasional discrepancies in detecting the correct versions of the Kernel, this was merely a superficial concern as it did not affect the detection of service versions. It is important to note that the artifact's scanning capabilities were limited to network-based scanning, but it effectively covered as much ground as possible in this regard.

Figure 11 shows the API output from a scan performed against a Microsoft Windows Server 2022 VM, using the script "vulscan" (not active by default), showing a list of found vulnerabilities from different databases.

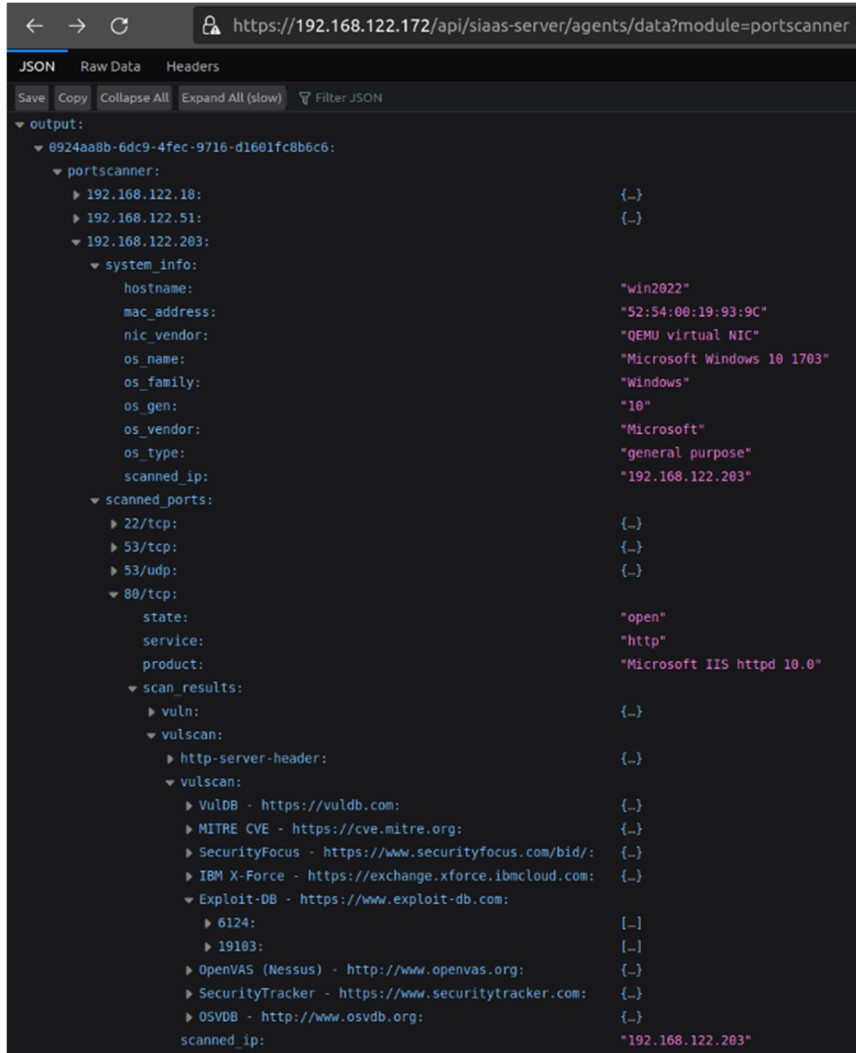

**Figure 11.** Scanning results for a Windows Server VM during local tests.

The implementation of this work is subject to an inherent limitation due to the type of scanner utilized. Specifically, the scanner employed is a "network-based" vulnerability scanner, which means it lacks the ability to access the internal patch level of the services operating on a host. Instead, it can only ascertain the upstream version displayed in the service's banner on a given port. Consequently, it is unable to determine whether a vulnerability has been resolved through the internal patching of the service. However, this issue could potentially be addressed by developing a module that can be installed on the target hosts. This suggestion is put forth as a future endeavor in Section 5.

### 4.1.3. Security Tests

Lastly, an additional examination was conducted to assess the security of the API. The user authentication process relies solely on basic HTTP authentication, which means that the login credentials are hardcoded in the host system (although they can be modified in the Server installation script). Consequently, only basic tests were performed to verify the proper implementation of the HTTPS protocol and simple HTTP authentication. A list of tests performed is given below:

1. Request against the server's API IP address (instead of the hostname), using the correct username/password, but without the endpoint's certificate to validate against (expected result—rejection);
2. Repeat, but explicitly using the client's flag to ignore certification validation (expected result—acceptance);
3. Request with proper username/password and CA bundle to validate against, but using IP address instead of hostname (expected result—rejection);
4. Repeat, but with correct hostname, and wrong username/password (expected result—rejection);
5. Repeat, but with correct user/name (expected result—acceptance);
6. Repeat, but using HTTP in the URL instead of HTTPS (expected result—client redirected from HTTP to HTTPS, and then accepted).

All these tests were successfully passed.

The raw outputs from all local tests can be consulted online (check the data availability links in the "Declarations" section).

### 4.2. User Tests

As said above, a total of four organizations and one freelancer tested and replied to the tester survey. Figure 12 shows a graphical distribution of the agreeableness replies obtained.

As illustrated in Figure 12, the vast majority of the responses fall within the categories of "agree" and "strongly agree", indicating a high level of overall satisfaction among all the testers.

Testers were requested to provide feedback on open-ended questions regarding problems encountered, and suggestions for improvement. The reported issues encompassed the vulnerability scanner's inability to identify backported vulnerabilities, which was already elaborated upon in the previous sub-section. Additionally, a well-known MongoDB limitation was identified in the agent's capacity to upload objects of larger sizes to the server, although this occurrence was infrequent. Another concern raised was the potential vulnerability of the agents to ARP spoofing, as their reliance on the ARP protocol for host discovery may potentially rend them susceptible. In terms of recommendations for future endeavors, the primary focus was on the development of a web interface. Furthermore, testers emphasized the importance of obtaining more comprehensive information regarding the target's operating system distribution versions and lifecycle. Lastly, containerization was proposed as an area for further exploration and improvement. All of this feedback has been incorporated as suggestions for future work in the subsequent section.

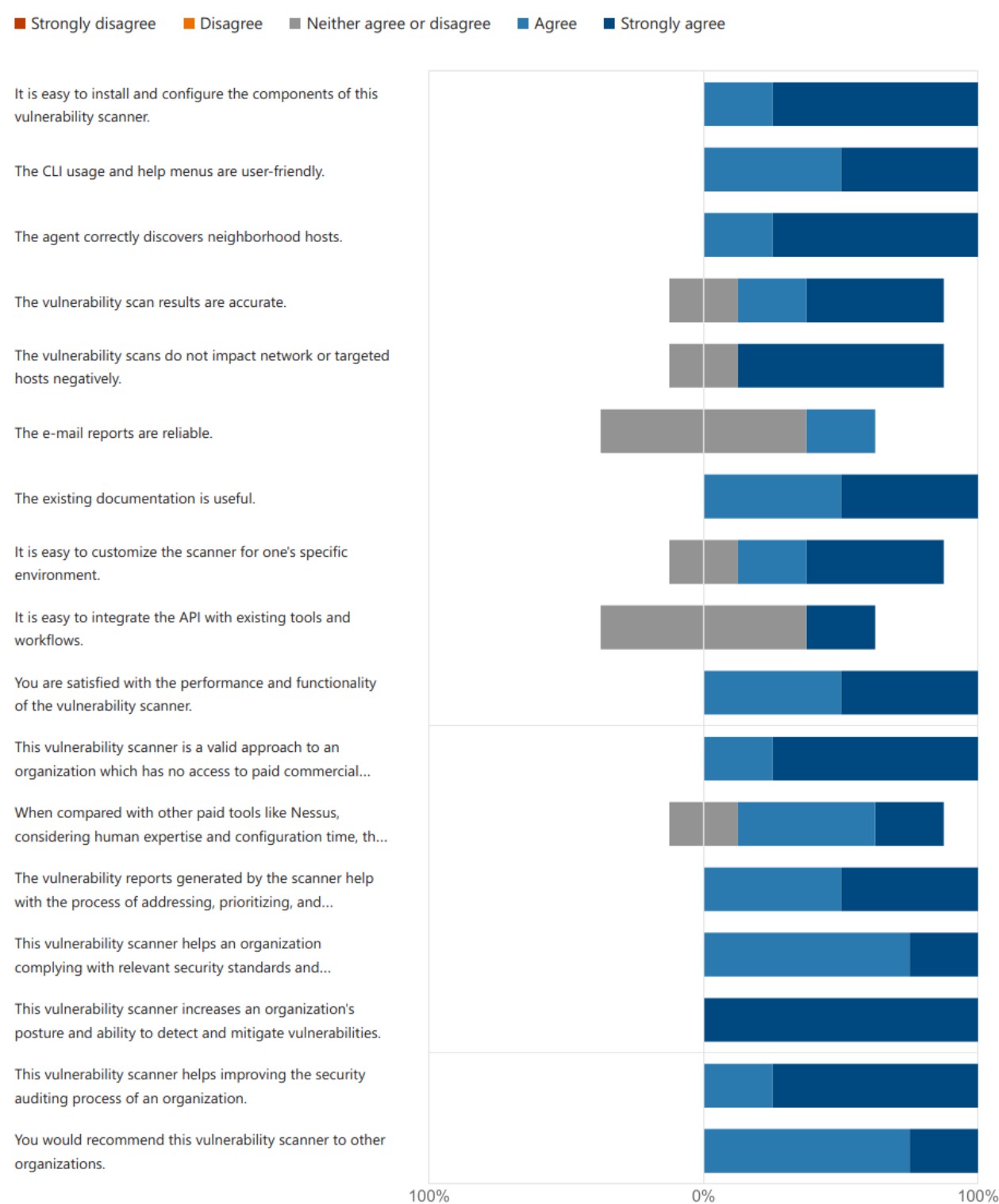

**Figure 12.** Graphical distribution of the agreeableness replies given by the testers.

The last inquiry was open-ended and revolved around concluding remarks. Each and every tester provided exceedingly positive feedback regarding their perception of the enhanced worth to their respective organizations. In fact, a few individuals even contemplated incorporating this valuable resource into their own set of tools.

To summarize, the artifact has been deemed user-friendly and positively impactful by external testers. Its implementation would enable organizations to streamline and enhance their cybersecurity measures effectively. By filtering out exploits, it assists in identifying

critical vulnerabilities that require immediate attention. Moreover, it has been established that this tool holds particular significance for organizations lacking paid solutions or dedicated cybersecurity experts.

## 5. Conclusions

The primary objective of this study was to create a vulnerability scanner that possessed user-friendly features, easy configuration, and scalability. Additionally, it aimed to ensure that the scanner was readily operational, compatible with inexpensive hardware, and accessible to the wider community.

In addition to the port scanning conducted by Nmap, which was selected for its simplicity, portability, efficiency, and seamless integration with Python, this study incorporates an automated discovery mechanism for identifying hosts in the vicinity. By utilizing this list of hosts, the port scanner can be effectively utilized without the need for extensive configuration, enabling system administrators lacking cybersecurity expertise to operate it with minimal effort. Nevertheless, the system's adaptability permits operators to configure a range of manual hosts, specify ports to scan, execute Nmap scripts, and make other customizations as desired.

The wide range of Nmap script categories available expands the scanner's capabilities beyond merely crosschecking services and versions against vulnerability databases. It also facilitates more advanced penetration testing, including assessments targeting specific services offered by particular vendors.

The work presented here follows an agent–server architecture. Consequently, the agents can be directly connected to internal LANs, bypassing the bureaucratic procedures and access restrictions typically associated with other scanners. This is particularly advantageous in data centers that prohibit inbound connections.

This work is designed to be highly modular, allowing an agent to operate independently from a server if required. The comprehensive API provided can be utilized by web interfaces developed by the community, enabling the potential for the creation of new features such as advanced graphing and reporting.

The artifact makes it possible to generate vulnerability reports and deliver them via email in a file format widely recognized as CSV. The level of detail in these reports can be customized, granting system administrators the ability to concentrate on addressing exploitable vulnerabilities that require immediate attention. Additionally, there is even the option of setting up an all-in-one (AIO) environment by installing both the agent and server in a single machine, or—as chosen by the testers of the artifact—a VM.

The examination of the artifact demonstrated its stability and reliability, even when used on older hardware that possesses fewer resources, provided that the appropriate configuration is in place. Despite the inherent limitations of the scanner type employed in this artifact, it effectively and accurately identifies hosts within the network, as well as their operating systems and services, along with any vulnerabilities present. The inclusion of multiple Nmap scripts enables the operator to select the most suitable option for a specific target host's operating system. Furthermore, the security-related tests, although minimalistic in nature, have indicated that the API implementation successfully prohibits unauthorized or untrusted connections.

Feedback obtained from external users, including those without a cybersecurity-specific background, has revealed that the artifact is user-friendly in terms of installation and operation. It accurately assesses the characteristics of the users' environments and provides a valuable vulnerability-related output, without causing any negative impact on the rest of the infrastructure. Additionally, all testers have expressed that the artifact proves to be effective in automating the security audit processes of organizations, especially in the absence of other (paid) options.

Based on the authors' examination of the current state-of-the-art solutions and their limitations (Section 2), one can conclude that the research outlined in this paper distinguishes itself from previous research efforts as, unlike any other solution, it consolidates all

of its provided features into a single artifact. Besides this, this artifact is also open-source, and can be easily installed and operated without the need for highly skilled personnel. Even when considering commercial/community solutions as well, the authors consider it can be confidently stated that there is currently no offering available that comprehensively addresses all the aforementioned aspects.

*Future Work*

The authors present the following recommendations for future research, drawing from their own observations and feedback received from testers.

In terms of UI/UX, a web frontend could serve as the interface between the server's API and the users. This frontend would have the potential to incorporate various features. Firstly, it could provide an AAA/SaaS multi-tenant cloud offering with a systematic approach to API security, ensuring the protection of sensitive data. Additionally, it could offer advanced graphical reporting, allowing users to visualize data in a more comprehensive manner. This frontend could also include stateful metadata information, enabling users to mark vulnerabilities as resolved and keep track of which vulnerabilities have already been reviewed. Moreover, it could notify users about new vulnerabilities found since their last visit. Furthermore, it could enhance e-mail reports, making them more informative and user-friendly. Another improvement could be the advanced scheduling of the agent's and server's modules, replacing the current fixed time interval approach. This would allow for more flexibility and efficiency in module execution. Additionally, a mobile application version of the web frontend could also be developed, to cater to users who prefer accessing the system through their mobile devices.

In the realm of AI/ML, an AI system could analyze the output of the API and determine appropriate courses of remediation. This AI system would leverage the data provided by the API to make informed decisions and suggest effective solutions. One possible use case would be suggesting which possible patches could be applied in a system to solve the existing vulnerabilities, using OS information, services running and their version, and the list of found vulnerabilities for each of these services. Additionally, outputs from the artifact could be used as inputs for ML algorithms that perform anomaly detection [29].

Regarding vulnerability scanning, the authors suggest the development of an optional agent-based software module that would be installed on target hosts. This module would provide additional information directly to the server, enabling a more detailed analysis. For example, it could gather information about backported vulnerability fixes for older versions of services, or obtain specific details about the OS distribution, version, and lifecycle.

Lastly, there is the possibility of expanding the capabilities of the agent by adding extra tools or modules. One suggestion is to incorporate specialized tools like Metasploit, which can be utilized for extensive pentesting. This would involve probing for exploits and running post-exploitable code, allowing for a more comprehensive assessment of system vulnerabilities.

**Author Contributions:** Conceptualization, J.P.S. and C.S.; methodology and investigation, J.P.S. and C.S.; software, J.P.S.; validation, J.P.S. and C.S.; writing—original draft preparation, J.P.S. and C.S.; writing—review and editing, J.P.S. and C.S.; supervision, C.S. All authors have read and agreed to the published version of the manuscript.

**Funding:** This research received no external funding.

**Data Availability Statement:** GitHub repository containing the source code, installation scripts, and "readme" files, of the Agent artifact: https://github.com/jpseara/siaas-agent (accessed 2 February 2024). GitHub repository containing the source code, installation scripts, and "readme" files, of the Server artifact: https://github.com/jpseara/siaas-server (accessed 2 February 2024). GitHub repository containing the source code, installation scripts, and "readme" files, of the CLI artifact: https://github.com/jpseara/siaas-cli (accessed 2 February 2024). GitHub repository containing complete configuration and API references, outputs of local tests, sample API outputs, and user survey form and response transcriptions: https://github.com/jpseara/siaas-research (accessed 2 February

2024). Original recordings of the user surveys can be requested by contacting the corresponding author(s) via e-mail.

**Acknowledgments:** We would like to thank all the testers of this work: Matt Golden—Trilio Data (USA); Ricardo Ramalho—Cybersecurity Behaviour and Automation at Altice Portugal (Portugal); Jorge Teixeira—VTXRM—Software Factory (Portugal); and David Negreira—Ubuntu community.

**Conflicts of Interest:** The authors declare no conflicts of interest.

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
