# Peer review of "Automation of System Security Vulnerabilities Detection Using Open-Source Software"

_electronics, doi:10.3390/electronics13050873_

Round 1

Reviewer 1 Report

Comments and Suggestions for Authors

Automating cybersecurity processes, particularly those pertaining to continuous vulnerability detection, are of importance. A cybersecurity vulnerability scanner was developed without any prior expertise from the operator. It was evaluated by IT companies and systems engineers. The scanner was user-friendly, and the results obtained could be utilized by the operator to proactively secure the systems involved in an automated manner.

The expression is not clear and sometimes misleading, for example, Compounding this issue is the scarcity of cybersecurity professionals, further exacerbating the problem.

In terms of conclusion, how might such research extend or improve the current state of knowledge in this field?

The references are too outdated and should be updated and introduced logically.

The authors provide the source code, installation scripts, and “readme” files in GitHub repository.

I think it can be accepted after minor revision.

Comments on the Quality of English Language

The expression is not clear and sometimes misleading, thus it can be polished.

Author Response

Hello!

Thank you so much for reviewing our work.

The paper was fully reviewed to accommodate your suggestions.

Inline our comments to your suggestions:

-> The expression is not clear and sometimes misleading, for example "Compounding this issue is the scarcity of cybersecurity professionals, further exacerbating the problem."

The entire paper was reviewed.

-> In terms of conclusion, how might such research extend or improve the current state of knowledge in this field?

This was explained in the end of section 2; but, to accomodate your observation, we have created an extra paragraph in the conclusions section, highlighting the same observations and making a back-reference to Section 2.

-> The references are too outdated and should be updated and introduced logically.

We have removed an old reference from 2013, and replaced with recent one from IBM.

Reviewer 2 Report

Comments and Suggestions for Authors

The authors emphasize the importance of automating cybersecurity processes, particularly continuous vulnerability detection, due to the scarcity of cybersecurity professionals and the specialized expertise required in the field. The paper proposes a user-friendly, freely available scanner that does not require prior cybersecurity knowledge from the operator. The effectiveness of the tool is evaluated through testing by IT companies and systems engineers. Overall, the manuscript addresses a relevant and timely topic and provides valuable insights into the automation of cybersecurity processes. The abstract provides a concise summary of the study, while the introduction effectively introduces the research problem and sets the context. The authors present a clear objective and contribution of their work. The related work section provides a brief overview of relevant studies, which could be further expanded to provide a comprehensive review of the existing literature. The subsequent sections on the design considerations, implementation, validation tests, and results are well-documented and provide sufficient detail.

Moreover, the manuscript would benefit from a more detailed description of the methodology used in developing the vulnerability scanner. Specifically, the authors should provide more information about the techniques and algorithms employed in the Network Discovery and Vulnerability Assessment modules. Additionally, it would be valuable to discuss the selection criteria for the validation tests and provide a clear description of the metrics used to evaluate the effectiveness of the scanner. The inclusion of quantitative data and statistical analysis would enhance the credibility of the results.

The authors highlight the significance of their research in addressing the challenges faced by organizations in cybersecurity. The provision of a user-friendly, cost-effective vulnerability scanning solution is a valuable contribution to the field. However, the manuscript could further emphasize the uniqueness and novelty of the proposed scanner by discussing how it compares to existing tools in terms of features, performance, and usability.

Finally, the manuscript lacks a section on future work, where the authors could discuss potential avenues for further research and development. Including this section would enhance the completeness of the study and provide guidance for future researchers interested in expanding upon the proposed vulnerability scanner.

Comments on the Quality of English Language

Some sentences are overly long and complex, making it difficult to grasp the intended meaning. It is recommended to revise these sentences for improved readability. Additionally, there are a few grammatical errors and typos that should be addressed through careful proofreading.

Author Response

Hello!

Thank you so much for reviewing our work.

The paper was fully reviewed to accommodate your suggestions.

Inline our comments to your suggestions:

-> Provide more information about the techniques and algorithms employed in the Network Discovery and Vulnerability Assessment modules.

Done.

-> Discuss the selection criteria for the validation tests and provide a clear description of the metrics used to evaluate the effectiveness of the scanner. The inclusion of quantitative data and statistical analysis would enhance the credibility of the results.

Done.

-> Further emphasize the uniqueness and novelty of the proposed scanner by discussing how it compares to existing tools in terms of features, performance, and usability.

This was explained in the end of section 2; but, to accommodate your observation, we have created an extra paragraph in the conclusions section, highlighting the same observations and making a back-reference to Section 2.

-> Lacks a section on future work, where the authors could discuss potential avenues for further research and development.

Created a new section "Future Work" under Conclusions.

-> Some sentences are overly long and complex, making it difficult to grasp the intended meaning. It is recommended to revise these sentences for improved readability. Additionally, there are a few grammatical errors and typos that should be addressed through careful proofreading.

The entire paper was reviewed.

Reviewer 3 Report

Comments and Suggestions for Authors

In this manuscript, the authors present an open-source vulnerability scanner that was developed and is quite easy to use. They present the requirements and the functionalities of the vulnerability scanner, illustrating its performance when utilized by non-experts (outside the cybersecurity domain). They also validated its performance both in a local laboratory and by external testers. According to the resulting outputs, the developed scanner was easy to use and can be effortlessly used by an operator to secure the ICT systems. The paper is in general interesting and well-written. However, it reads as a technical report and, in my view, major revisions are required before publication:

·       You mention in the introduction that the vulnerability scanner consists of 3 modules (network discovery, vulnerability assessment, reporting). Please elaborate on the implementation section and provide the connection between these modules (preferably graphically).

·       You should clearly describe the vulnerability detection process in a separate section.

·       What is the feature of the scanner that provides intelligence? Is the process based on an intelligent algorithm? Please elaborate. If not, the title of the manuscript is misleading. It would be good to mention that the scanner can be extended to include ML functionalities (for instance anomaly detection) in order to pro-actively detect vulnerabilities, classify and assess them. Relevant references for similar ML algorithms are:

1.       "An unsupervised deep learning model for early network traffic anomaly detection." IEEE Access 8 (2020): 30387-30399.

2.       "Intelligent Mission Critical Services over Beyond 5G Networks: Control Loop and Proactive Overload Detection." 2023 International Conference on Smart Applications, Communications and Networking (SmartNets). IEEE, 2023.

3.       "Evaluation of machine learning algorithms for anomaly detection." 2020 international conference on cyber security and protection of digital services (cyber security). IEEE, 2020.

Author Response

Hello!

Thank you so much for reviewing our work.

The paper was fully reviewed to accommodate your suggestions.

Inline our comments to your suggestions:

-> You mention in the introduction that the vulnerability scanner consists of 3 modules (network discovery, vulnerability assessment, reporting). Please elaborate on the implementation section and provide the connection between these modules (preferably graphically).

Done.

-> You should clearly describe the vulnerability detection process in a separate section.

Done (created a sub-section in Implementation).

-> What is the feature of the scanner that provides intelligence? Is the process based on an intelligent algorithm? Please elaborate. If not, the title of the manuscript is misleading. It would be good to mention that the scanner can be extended to include ML functionalities (for instance anomaly detection) in order to pro-actively detect vulnerabilities, classify and assess them. Relevant references for similar ML algorithms are:

Good point. The "inteligence" part was meant to be, at the beginning of this work, the fact that the artifact was smart enough to find all hosts in the neighborhood and automatically all the related ports. But it's more accurate if the word "intelligence" is not part of the title, as the artifact does not have any intelligence per se. So it was removed.

Regarding the second part of your comment, it is explained in the 5.1 Future Work section that this work could be extended to include ML functionalities.

Finally, regarding your reference suggestions (thanks for those), here goes our comments:

1. "An unsupervised deep learning model for early network traffic anomaly detection." IEEE Access 8 (2020): 30387-30399.

(The authors developed an deep learning model that performs anomaly detection systems that is focused on packet analysis; not the scope of this artifact.)

2. "Intelligent Mission Critical Services over Beyond 5G Networks: Control Loop and Proactive Overload Detection." 2023 International Conference on Smart Applications, Communications and Networking (SmartNets). IEEE, 2023.

(Mostly focused on 5G networks which is not the main focus of the artifact.)

3. "Evaluation of machine learning algorithms for anomaly detection." 2020 international conference on cyber security and protection of digital services (cyber security). IEEE, 2020.

This article is interesting, as the API outputs of our artifact could be fed into such a ML algorithm. Added.

Round 2

Reviewer 3 Report

Comments and Suggestions for Authors

In my view, the authors have responded to the comments raised by the reviewers, significantly enhancing the quality of their manuscript.